# Implementing Health Warnings on Alcoholic Beverages: On the Leading Role of Countries of the Commonwealth of Independent States

**DOI:** 10.3390/ijerph17218205

**Published:** 2020-11-06

**Authors:** Maria Neufeld, Carina Ferreira-Borges, Jürgen Rehm

**Affiliations:** 1WHO European Office for Prevention and Control of Noncommunicable Diseases, Leontyevsky Pereulok 9, 125009 Moscow, Russia; ferreiraborgesc@who.int (C.F.-B.); jtrehm@gmail.com (J.R.); 2Institute for Clinical Psychology and Psychotherapy, TU Dresden, Chemnitzer Str. 46, 01187 Dresden, Germany; 3Institute for Mental Health Policy Research, Centre for Addiction and Mental Health (CAMH), 33 Ursula Franklin Street, Toronto, ON M5S 2S1, Canada; 4Campbell Family Mental Health Research Institute, CAMH, 250 College Street, Toronto, ON M5T 1R8, Canada; 5Institute of Medical Science (IMS), University of Toronto, Medical Sciences Building, 1 King’s College Circle, Room 2374, Toronto, ON M5S 1A8, Canada; 6Department of Psychiatry, University of Toronto, 250 College Street, 8th Floor, Toronto, ON M5T 1R8, Canada; 7Dalla Lana School of Public Health, University of Toronto, 155 College Street, 6th Floor, Toronto, ON M5T 3M7, Canada; 8Department of International Health Projects, Institute for Leadership and Health Management, Sechenov First Moscow State Medical University (Sechenov University), Alexander Solzhenitsyn Street 28/1, 109004 Moscow, Russia

**Keywords:** alcohol, alcohol policy, alcohol warning, consumer health information, Commonwealth of Independent States, Eurasian Economic Union, food labeling, food safety, health promotion, warning labels

## Abstract

Despite being a psychoactive substance and having a major impact on health, alcohol has to date escaped the required labeling regulations for either psychoactive substances or food. The vast majority of the countries in the WHO European Region have stricter labeling requirements for bottled water and health warning provisions for over-the-counter medications than for alcoholic beverages. However, more progress in implementing health warnings has been made in the eastern part of the WHO European Region, largely because of the recent technical regulation put in place by the newly formed Eurasian Economic Union. The present contribution provides an overview of the existing legislation regarding the placement of alcohol health warnings on advertisements and labels on alcohol containers in the countries of the Commonwealth of Independent States (CIS; Armenia, Azerbaijan, Belarus, Kazakhstan, Kyrgyzstan, Moldova, Russia, Tajikistan, Turkmenistan, and Uzbekistan) and discusses their potential gaps and shortfalls. It also reviews the evolution of the Eurasian Economic Union Technical Regulation 047/2018, which is, to date, the only international document to impose binding provisions on alcohol labeling. The technical regulation’s developmental process demonstrates how the comprehensive messages and strong requirements for health warnings that were suggested initially were watered down during the consultation process.

## 1. Introduction

As of October 2020, the vast majority of countries in the WHO European Region have stricter labeling requirements for bottled water, food products, and over-the-counter medications than for alcoholic beverages because alcoholic beverages, as a specific category, have evaded the international labeling standards required for psychoactive substances and food [1,2]. Both the WHO Global Strategy to Reduce the Harmful Use of Alcohol and the WHO European Action Plan to Reduce the Harmful Use of Alcohol call for the “provision of consumer information about, and labeling of, alcoholic beverages to indicate the harm related to alcohol.” The present contribution will only deal with the latter part of this provision for health warning labels.

The evidence of labeling effectiveness on changing consumer behaviors in relation to smoking and nutrition has been accumulating over the past 20 years, but for alcohol specifically, empirical studies have only recently been available. The latest series of publications on alcohol health warnings and their impact on risk perception, readiness to reduce drinking, and actual behavioral changes highlight that health warnings on alcohol labels are an important but yet not very well explored tool for decreasing alcohol consumption and the associated harms [3,4,5]. The quasi-experimental study in Yukon, Canada, clearly showed that the introduction of health warnings and further information on alcoholic beverages can result in reduced per capita consumption in a community, with the largest impact being observed in heavy drinkers, i.e., the individuals who see the health warnings most frequently and are at the greatest risk of experiencing alcohol-attributable harms [3].

There is an obvious parallel to tobacco regulations and their successes over time in halting the global tobacco epidemic [6]. After the adoption of the World Health Organization Framework Convention on Tobacco Control (WHO FCTC) in 2003, large, clear, and visible health warnings on tobacco packaging and advertisements became a standard practice to inform consumers about the risks that they were taking when consuming these products [7]. Research has shown that tobacco health warnings, especially pictorial pack warnings, are effective at increasing risk awareness, including cancer risk, and reducing tobacco use [8,9,10]. For instance, a systematic review of 32 studies in 20 countries with more than 800,000 participants demonstrated that cigarette pack warnings were associated with increased knowledge, attempts at quitting, and decreased smoking [11].

Alcohol was classified by the International Agency for Research on Cancer (IARC) as a group 1 carcinogen 30 years ago, with tobacco smoking, tobacco smoke, smokeless tobacco, and second-hand tobacco smoke being classified in the same group as agents that are carcinogenic to humans [12]. Yet, this causal relationship between exposure and cancer is much less commonly known for alcohol than for tobacco. Introducing cancer-specific health warnings for alcohol might be a cost-effective tool for raising awareness of the associated risks and replicating the success of tobacco health warnings. The results from the latest round of the Global Drug Survey, encompassing more than 75,000 participants from 29 countries, indicated that the link between cancer and drinking was the least well known but encouraged almost 40% of current drinkers to reduce their intake [13].

Adding warning labels to alcohol containers that link consumption to the risk of harms, for instance, the risk of alcohol-attributable cancer, has been shown to have great potential for not only increasing knowledge and awareness but also for increasing public receptiveness to alcohol control policies [14].

Finally, health warnings need to be considered not only as a policy tool for decreasing consumption and harms but also as a mandatory part of product labeling to provide consumers with information on a product’s ingredients, nutritional and caloric values, and potential harms. The provision of health warnings on alcoholic beverages to inform their customers is therefore the producer’s obligation [15].

Unlike for tobacco, health warnings as part of a standard labeling practice are poorly implemented in many countries around the world and most alcohol consumers are not aware of the associated risks. The latest WHO Health Evidence Network (HEN) synthesis report on alcohol labeling across the WHO European Region highlighted that alcohol labeling, let alone health warnings, is not mandatory in many countries in the region [2]. The report also revealed a large gap at the sub-regional level; while most European Union (EU) member states were lacking any kind of regulation and less than a third had any provisions regarding health warnings placed on containers, non-EU countries tended to implement stricter rules on alcohol labeling and health information and tended to meet the recommendations made in the WHO’s discussion paper on policy options for alcohol labeling [16]. Although the HEN report is the first systematic study that covers alcohol-labeling practices in the WHO European Region and features various country examples, it does not provide a more comprehensive in-depth analysis of the specific alcohol health warnings implemented in many of the non-EU countries.

The present article takes this report as a point of departure and provides an overview of the current state of implementation of health warning labels on alcoholic beverages in the 10 countries of the Commonwealth of Independent States (CIS; Armenia, Azerbaijan, Belarus, Kazakhstan, Kyrgyzstan, Moldova, Russia, Tajikistan, Turkmenistan, and Uzbekistan) and discusses specific issues related to the content of the health warnings. The article also discusses the development of the technical regulations on the safety of alcoholic beverages in the Eurasian Economic Union (EAEU), which is an economic and political union between Armenia, Belarus, Kazakhstan, Kyrgyzstan, and Russia, and its specific guidelines on the requirements for health warnings on alcohol labels. Although this article clearly focuses on the implementation of health warning labels on alcoholic beverages and discusses some countries’ examples in greater detail, it also briefly highlights two related areas, namely, the provision of other information printed on the label (i.e., the ingredient list) and the provision of health warnings in alcohol advertisements.

In most CIS countries, alcohol health warnings on alcohol containers and alcohol advertisements are in place or alcohol marketing is either prohibited outright or is substantially limited. So far, however, not a single study has been carried out that documents the link between the implementation of health warnings and attitudes toward drinking, awareness of the risks, and/or behavioral change. While we do not provide any analyses on the impact or effectiveness of health warnings in the CIS in this contribution, this article can be seen as the first thorough overview of the precise health warning requirements in the CIS and the associated regulatory frameworks as a point of departure toward more comprehensive research in these countries.

## 2. Materials and Methods

A qualitative policy document analysis of national legislation and other regulatory documents was carried out for the 10 countries of the CIS (all member states of the CIS and Turkmenistan as an associate member state). The documents were mainly searched for in the “Legislation of the CIS countries” database [17] and legislation related to advertising were additionally searched for on the website of the Russian advertising agency “Geography,” which contains a database of all alcohol advertising laws in the CIS in Russian languages [18]. The searches were carried out in Russian by the first author using the search phrases “alcohol,” “alcoholic beverage,” “warning,” and “labeling.” For each country, the respective main law regulating alcohol was searched for using these phrases, although not all countries had such a law in place.

The present analysis was also informed by a workshop on alcohol control policies in the CIS that took place in December 2019 in Moscow, where delegates provided input on the current state of alcohol policy implementation and presented examples of alcohol health warnings in their countries [19]. Technical regulations on alcohol labeling from the Eurasian Economic Union were retrieved from its official website [20].

## 3. Results

### 3.1. National and Supranational Regulations of Alcohol Health Warnings in the CIS

Almost all CIS countries have national regulations in place that require health warnings on labels affixed to alcohol containers or to be incorporated into the text in advertisements wherever alcohol advertising is allowed. Five of the 10 CIS countries are also member states of the Eurasian Economic Union, which has its own regulatory standards for food and alcohol labeling that also mandate health warnings on the packaging of alcoholic beverages (for an overview of the countries discussed and the national/supranational health warning provisions, see Figure 1).

The Eurasian Economic Union’s technical regulation (TR EAEU 047/2018), “On safety of alcoholic beverages,” applies to all types of alcoholic beverages intended for human use in the territory of the EAEU member states (for the distinction between alcohol for human use and so-called surrogate alcohol, see [21]). It mandates the provision of an ingredients list, a health information message, and an additional message of a “recommendatory nature” [2,22]. The regulation specifies that the health information message: “Excessive consumption of alcohol is harmful to your health” should be a “contrasting” message, written in capital letters in an easy-to-read font of the largest possible size and to occupy at least 10% of the label’s surface (front or back label or anywhere else on the package surface). The additional recommendatory message: “Alcohol use is not recommended for persons under the age of 18, pregnant and breastfeeding women, as well as persons with diseases of the nervous system and internal organs” should be featured on the label as well, but the exact font, size and other design features are not specified in the regulation. For Kazakhstan, the message should read as follows, in accordance with the national minimum drinking age: “Alcohol use is not recommended for persons under the age of 21, pregnant and breastfeeding women, as well as persons with diseases of the nervous system, kidneys, liver and digestive tract.” Both, the health warning and the additional message have to be provided in the local national language and in Russian.

This technical regulation was adopted by the decision of the Council of the Eurasian Economic Commission of 5 December 2018 N 98 [22] and, following the established rules of the Eurasian Economic Union, it will enter into force two years later, on 9 January 2021, and will supersede any previous national regulations [23]. During this two-year period, member states have to ensure that their national products are in line with the regulation but they are free to use alcohol labels that are in line with national regulations in the interim period. Beginning on 10 January 2021, all alcohol producers will have to comply with the TR EAEU 047/2018 or their licenses will be suspended.

As for national regulations on alcohol health warnings in the CIS, an overview is provided in Table 1.

Almost all CIS countries have national and supranational legislation in place that require health warnings either on alcohol advertisements, alcoholic beverage containers, or both. Out of the 10 CIS countries, Tajikistan is the only country that has no provisions in place in relation to alcohol health warnings.

Four countries (Kazakhstan, Tajikistan, Turkmenistan, and Uzbekistan) have a total ban on alcohol and tobacco advertising across all media in place, which is why the issue of health warnings in alcohol ads does not formally apply to them. In Russia and Belarus, alcohol advertisements need to be accompanied by the same health warning as on the alcohol labels (“Excessive alcohol consumption is harmful to your health”), which has to occupy at least 10% of the advertisement space. The advertising laws of Azerbaijan and Kyrgyzstan also impose health warnings but do not specify requirements for the content or size. Moldova and Armenia have no regulations in place in relation to alcohol advertising and health warnings, meaning that out of all CIS countries, these are the only two countries where alcohol advertising is allowed and does not have to be accompanied by a health warning.

### 3.2. Examples of Health Warnings on Labels of Alcoholic Beverage Containers in the CIS

Armenia is the only country that has no national regulation on health warnings in place (neither on advertisements nor alcohol labels), but since it is an EAEU member state, the rules of the above-mentioned Technical Regulation EAEU 047/2018 must be followed. Because of the two-year “grace period” for producers to adapt to the new technical regulation, alcoholic beverages in Armenia can now be found with and without the health warning (see Figure 2).

The same is true for Kyrgyzstan, which has had no national legislation on alcohol health warnings but now has to follow the EAEU regulations. Belarus, Russia, and Kazakhstan already had national regulations in place, and Kazakhstan only recently repealed its national resolution on health warnings since they did not substantially differ from the EAEU provisions and the producers were already adhering to the international rules.

None of the CIS countries has cancer-specific health warnings, either on advertisements or on container labels. Most of the reviewed national regulations do not feature specific guidelines for the size, content, or design of the health warnings, such as the font or the need for a contrasting background.

None of the reviewed health warnings were fully aligned with the main principles detailed in the WHO discussion document on policy options for alcohol labeling [16]. As for the overall design options, the document lists the following standards: (1) the labels are to be placed in a standard location; (2) the size is to be determined as a minimum percentage of the container size; (3) the use of rotating messages with sufficient vividness and strength to attract consumers; (4) the text is clearly separated from other information on the label (for example, placed in boxes with thick borders); (5) the text printed in capital letters and bold type, where its size is the same as for all other information provided on the container; (6) the text appears on a contrasting background (for example, red text on white); (7) the text is written in the official language(s) of the country in which the product is sold; (8) images are to be informational in style and taken from ongoing educational campaigns; (9) public health bodies are to advise on the content of the messages. Regarding the specific content of the health messages used, the WHO suggests four message components to be considered when developing the text of the health warning: (1) a signal word to attract attention, (2) an identification of the problem, (3) an explanation of the consequences if exposed to the problem, and (4) instructions for avoiding the problem [16].

As already shown by the example of the Armenian health warnings (Figure 2), the technical regulations of the EAEU are not fully in line with these suggested policy options as it does not, for instance, require a clear visual separation between the health warning and other information printed on the label. This allows the alcohol producers to self-regulate, allowing them to introduce the health warning in the middle of the label where it does not attract much attention, as shown in the Armenian example. Another example from Belarus (Figure 3) also shows how the health warning can be designed in the same color scheme and style as the rest of the label, so it does not stand out too much, despite its large size, bold font, and text in capital letters.

Another example from Uzbekistan, which is not part of the EAEU and has its own national regulations on alcohol health warnings in place, shows how the “40% of label size” requirement can remain largely unnoticed on the label because there is no separation between the health warning and the other visual elements, allowing the warning to merge with the overall label design (Figure 4).

Uzbekistan is also free to use a graphical health warning and/or a pictogram on the label, but to our knowledge, this option has never been used and a more comprehensive study with a systematic assessment of health warnings on alcohol containers would be needed to assess the implementation of graphic health warnings in that country. Besides Uzbekistan, Moldova is the only country that has an explicit regulation on pictograms in place. Instead of a text message, alcohol labels must feature two pictograms to warn about the harms of alcohol use by vulnerable population groups, namely, pregnant women and youth under the age of 18.

Turkmenistan is the only country to provide clear rules on the contrast between the health warning and the label background by stating that the message “Alcohol is harmful to your health!” must be printed in black capital letters on a white background. Although the legislation does not outline any further requirements for the separation between the health warning and the background, some producers place the health warning in a box with distinct borders, which results in labels with very distinct and attention-grabbing health warnings (see example in Figure 5).

### 3.3. The Technical Regulation “On Safety of Alcoholic Beverages” of the Eurasian Economic Union

As for the EAEU technical regulation “On safety of alcoholic beverages” (TR EAEU 047/2018), the history of its development shows that the initial regulation draft was much stronger and much more aligned with the WHO principles of design and content (for a comparison of the main differences between the different draft versions of the regulation and the adopted version, see Table 2).

The process of drafting the technical regulation was initiated in 2011 by the Russian Federation. The first and initial draft suggested the more general health warning “Alcohol is harmful to your health” with a size of at least 20% of the label’s surface, which should be printed in black capital letters on a white background, in a bold, clear, easy-to-read font of the largest possible size with line spacing not exceeding the font height [30]. The document was presented on the official website of the Ministry of Economic Development for public consultations, the outcomes of which were made public soon thereafter in the resulting resolution [31], which was more aligned with the alcohol industry’s submissions rather than those of the public health stakeholders. Overall, the document followed more of an economic line of argument that mirrored the producers’ perspectives [32]. For instance, criticism was raised stating that the requirements for the placement of a warning label were not formulated accurately enough and did not adequately consider the variety of types of packaging used for alcoholic beverages. It was further noted that the procedures for applying a warning label on consumer packaging that already had several labels were not clear and the proposed size of the health warning was seen as “excessive” since it could compromise the placement of other information that is required by law. At the same time, the suggested shorter message “Alcohol is harmful to your health” was rejected by an argument stating that there was no justification provided for why the current (and much longer) message “Excessive alcohol consumption is harmful to your health” was ineffective.

The feedback also called for an elimination of the requirement of a colored font and background, arguing that this would interfere with the branded design of alcohol labels, which is considered to be the intellectual property of producers. The specific paragraph on the resolution concluded that the introduction of health warnings in the proposed form might “seriously reduce” alcohol imports to the Russian Federation and “influence the integration processes on the entry of the Russian Federation into the World Trade Organization” [32].

During this discussion, some experts mentioned that text warnings are not enough and that graphic warnings of damaged organs should be introduced, following the example of tobacco [33].

A second draft was disseminated to other countries of the Customs Union for further consultations through the World Trade Organization in 2015 and discussed up until 2018 [34,35], when it was adopted in a weaker form from a public health perspective. For instance, the required size of the health warning was reduced by half and the proposed health information message that alcohol use was not recommended to people under the age of 21 could only be adopted by Kazakhstan to be in line with the national minimum drinking age, while for all other countries, it was reduced to the age of 18 (for the main differences, see Table 2).

## 4. Discussion

To our knowledge, this is the first review that summarizes the state of the implementation of health warnings in CIS countries. Most of the literature on alcohol labeling and health argues that the provision of health warnings on alcohol labels has the potential to increase awareness of the health and social harms stemming from alcohol consumption, but this seems to largely depend on the specific content and design of the health warnings, as has been seen in the case of tobacco. The results of this short review on health warnings implementation in CIS countries support this overall rationale and highlight the importance of both the content and the design of alcohol labels. Although we do not address the provision of health warnings in alcohol advertisements as such, we nevertheless have included this into the review to highlight that in the majority of countries, alcohol advertising is either prohibited or must be accompanied by a health warning [36,37,38,39]. From a public health perspective, as well as an implementation perspective, a complete ban on alcohol marketing and promotion appears to be more effective and easier to enforce than partial restrictions and/or the provision of health warnings on alcohol advertisements. However, having mandatory health warnings on alcohol advertisements is better than not regulating alcohol advertisements at all, and experimental findings, as well as examples from other countries, show that health warnings increase risk perception and reduce advertising’s effects [40]. Overall, it should be noted that compared with other countries in the WHO European Region, CIS countries have relatively strict provisions on alcohol advertising in general [36,41,42].

The literature on alcohol health warnings in the CIS does not exist and we could not identify any historic sources that would document the Soviet provisions on health warnings on alcohol labels and their development over time. It is possible that health warnings were introduced as part of the various Soviet anti-alcohol campaigns that were aimed at reducing alcohol-related harms in the Soviet Union [43]. Our attempt to conduct a review of Russian literature on this topic failed because the searches yielded no results, with the exception of the original regulatory documents we have reviewed and news articles announcing the regulatory documents. For instance, different news outlets reported that there were attempts to introduce graphic health warnings on alcohol labels in 2018 in Azerbaijan [44], Kyrgyzstan [45], and Russia [46], which were brought forward as legal initiatives by parliament members or other politicians. In the case of Russia, the initiative to introduce graphic health warnings also included a suggestion to implement a more direct and urgent message of “Alcohol kills” in order to replace the message “Excessive alcohol consumption is harmful to your health.”

However, some Russian-language studies were identified that have at least in part addressed health warnings as a viable tool to inform consumers’ choices. For instance, Balashova and colleagues state, as part of a larger research project on fetal alcohol spectrum disorders in Russia, that printed materials, including health warnings on alcohol labels, have effects on pregnant women [47]. A literature review on the prevention of drink driving in Russia reports on a corporate social responsibility campaign where an international transport company and a local restaurant chain issued a limited collection of homemade wines in Moscow in 2018, the labels of which depicted scenes of car accidents [48]. Another literature review on the advertisement of alcohol and tobacco products in Russia draws attention to the already discussed issue of the health message “Excessive alcohol consumption is harmful to your health” [49]. The authors reported a case in which this message was manipulated and used to promote alcohol. A printed advertisement of vodka in a journal featured the following health warning “Only excessive alcohol consumption is harmful to your health,” which occupied only 4% of the advertising space instead of the designated 10%. The courts that dealt with this case concluded that by adding the word “only,” the advertisement implied that any other alcohol consumption was harmless to human health, and they, therefore, found the advertisement to be inappropriate. A study that looked at the issue of illegal alcohol sales in Russia and the sale of alcohol to minors briefly mentioned that the presence of health warnings was routinely checked for by authorities when seizing illegally produced alcohol, and when they composed a docket of seized goods, but noted that health warnings were usually in place on the counterfeited alcohols [50].

The two regulations most aligned with the WHO-suggested principles of alcohol labeling in terms of the design and content of the health warnings are the technical regulation of the EAEU and the regulations from Turkmenistan. While Turkmenistan is the only country to use a direct warning of “Alcohol is harmful to your health!” on its labels, other countries use the term “excessive alcohol consumption,” or, as in the case of Uzbekistan, the clinical term “alcohol abuse” instead. While the latter is an obvious tautology because “alcohol abuse” is already the name of an alcohol use disorder, per the DSM–IV, the term “excessive alcohol consumption” seems equally problematic, as it can be read in a way to suggest that there is a way to consume alcohol that would not have health consequences. This message has been criticized by public health experts from CIS countries [19]. For instance, the Ministry of Health of Belarus declared in a public letter in relation to this health warning message that it “consistently adheres to the position that any alcohol consumption is harmful to human health” [24]. The evolution of the TR EAEU 047/2018 showed that the initial suggestion to introduce the explicit health warning “Alcohol is harmful to your health!” was denied after the draft went through a public consultation process, involving among other stakeholders, including alcohol producers (see Table 2).

As indicated above, although the TR EAEU 047/2018 imposes the specific rules for the content of the warning messages and their size such that individual countries may no longer introduce any other messages at the national level, the legislation does not explicitly forbid the use of graphical health warnings on the label. However, such additional national labels would have to be examined by the World Trade Organization Committee on Technical Barriers to Trade [51] which aims to ensure that technical regulations, standards, and procedures are non-discriminatory and do not create unnecessary obstacles to trade. To our knowledge, the only such examination of suggested new pictorial labels by the government of Thailand failed because of objections from other member states of the World Trade Organization [51] In addition, the World Trade Organization stipulated general conditions to Thailand as minimal conditions [52]. As the example of tobacco pictorial legislation shows, the more evidence that exists on the exact impact of graphical messages, the higher the chances it can pass.

Therefore, while countries may no longer use any other health messages than those imposed by the regulation, the TR EAEU 047/2018 does not forbid countries to introduce graphic health warnings to illustrate alcohol-induced harms if they wish to do so, for instance, graphic warnings of alcohol-attributable cancers. However, such labels would have to undergo a test by the World Trade Organization Committee on Technical Barriers to Trade [51], and, to our knowledge, the only such examination of suggested new pictorial labels for alcohol, brought forth by Thailand, failed [51].

Moreover, changes to the TR EAEU 047/2018 are not the only possible way to introduce comprehensive health warnings on the harms of alcohol. For instance, the European Union lacks any regulatory document imposing health warnings. However, the European Commission is currently working on its “Europe Beating Cancer Plan,” [53] and it remains to be seen whether cancer-specific health warnings for alcohol could be considered as a policy option as part of this plan. The policy reforms that come with the Eurasian Economic Community provide various opportunities to introduce public health policies, including graphic health warnings for alcohol.

## 5. Conclusions

Considering both national and supra-national regulatory frameworks, CIS countries are much more advanced in implementing health warnings on alcoholic beverages, as well as in alcohol advertisements, than the rest of the WHO European Region. However, as has been shown by some examples of alcohol labels from these countries, it is not enough to have legislation in place that generally impose health warnings; it is equally important to have clear public health guidance on the content and form of the health warnings such that they attract the consumer’s attention and deliver clear risk warnings. Monitoring compliance with regulatory frameworks is also crucial in this regard.

Despite its great importance as the first international document to require health warnings on alcoholic beverages, the development of the EAEU technical regulation on alcohol labeling shows that not all its potential has been used in order to have clear and impactful health warning label design and messaging. The opportunity to introduce health warnings that would be more in line with international guidance from public health experts seems lost at the moment because the technical regulations enter into force as it currently stands in just a few months. This might be due to the lack of knowledge and resources on the part of the developers on the specific issues related to alcohol labeling, but it may also be due to industry interference, which succeeded in changing the final wording and the sizes of the labels during the consultation process with the public, as noted earlier. However, the EAEU regulation specifies minimum joint requirements for all member states, and as long as national regulations are in line with these, they can go beyond these EAEU minimum requirements. For instance, graphical alcohol labels could be suggested by a member country; however, they would have to undergo the processes stipulated by the World Trade Organization (see below; for details of experiences of the only country that tried to legislate graphical alcohol labels, see [51,54,55]). With the exception of Armenia, EAEU countries have strong alcohol policies in place and overall declines in alcohol consumption have been observed in all five member states [36]. These countries can further draw on the full potential of the EAEU harmonization process and establish high alcohol excise rates across all countries (something that is currently under discussion and development) and explore further possibilities of the joint economic space to raise awareness of alcohol as a risk factor for consumers and thereby reduce the harms associated with it. So far, the harmonization of the legislation of the EAEU countries has mainly been driven by economic factors because of the nature of this trade union, with specific roadmaps for action being developed mostly for the energy and infrastructure sectors. It is crucial that public health interests and arguments also guide the debates and the policy reforms, and that alcohol gets the attention it deserves in this process, given the role it plays in shaping the demographic and mortality trends in these countries.

The results presented above, and specifically the discussion of some examples of labels on alcoholic beverages, highlights yet again that it is not only important that there is a binding regulation on the provision of health warnings as such but also specific guidance on their design, size, and content [2,16]. Without this, producers will find creative ways to circumvent the regulations. Finally, the provision of health warnings on alcoholic beverages should be part of a comprehensive approach to inform individuals, communities, and societies about the alcohol-related harms and facilitate behavioral change. Health warnings on labels should not be considered stand-alone measures to raise awareness of general and specific risks associated with alcohol. They must be linked to specific guidelines, screening, prevention, and intervention programs within the healthcare system and community settings, as well as an overall regulatory framework that recognizes alcohol as “no ordinary commodity” but as a major risk factor and determinant of ill health.

## Figures and Tables

**Figure 1 ijerph-17-08205-f001:**
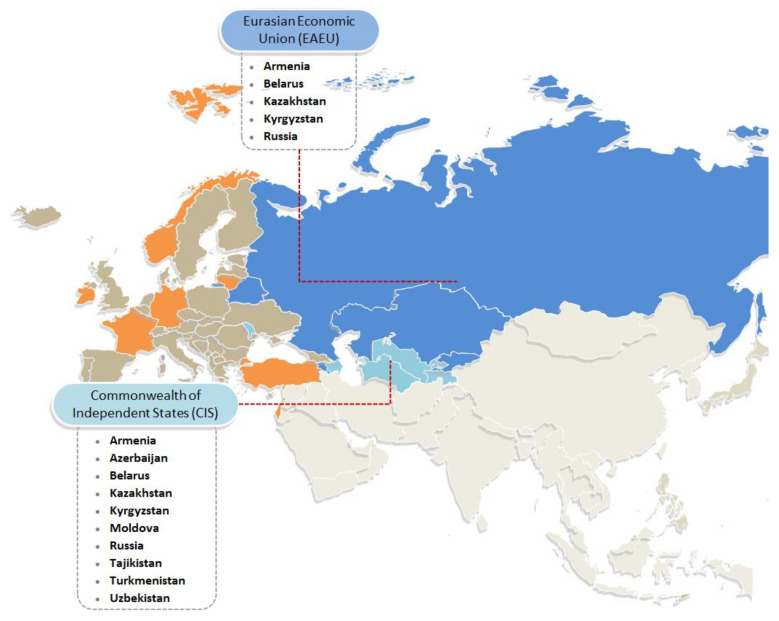
An overview of the implementation of health warnings in the CIS and the rest of the World Health Organization (WHO) European Region. In blue: CIS countries that are also member states of the EAEU where the Technical Regulation 047/2018 will soon come into force, superseding any previous national regulations. In turquoise: other CIS countries (for an overview of their national legislation on health warnings, see Table 1). In orange: other countries of the WHO European Region that require health warnings to be placed on alcohol containers. Adapted from: World Health Organization Regional Office for Europe [2].

**Figure 2 ijerph-17-08205-f002:**
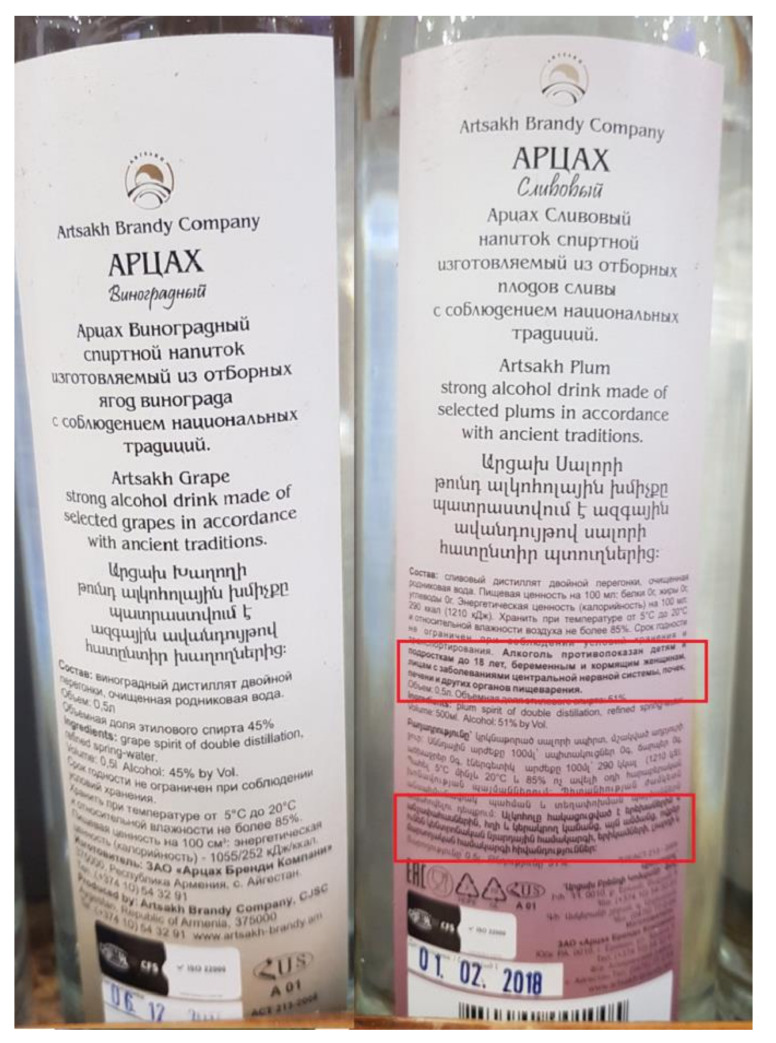
An example of an Armenian spirits package with and without the EAEU recommendatory information message. Left: a container that follows national regulations, featuring information on the ingredients only (in Russian and English). Right: a container that follows the international EAEU regulations, featuring the recommendatory message: “Alcohol use is not recommended for persons under the age of 18, pregnant and breastfeeding women, as well as persons with diseases of the nervous system and internal organs” in bold letters (in Russian and in Armenian). The recommendations are framed in red for better readability and seem to occupy at least 10% of the label’s surface). However, the health warning itself “Excessive alcohol consumption is harmful to your health” is not seen on the label and is most likely printed elsewhere on the container.

**Figure 3 ijerph-17-08205-f003:**
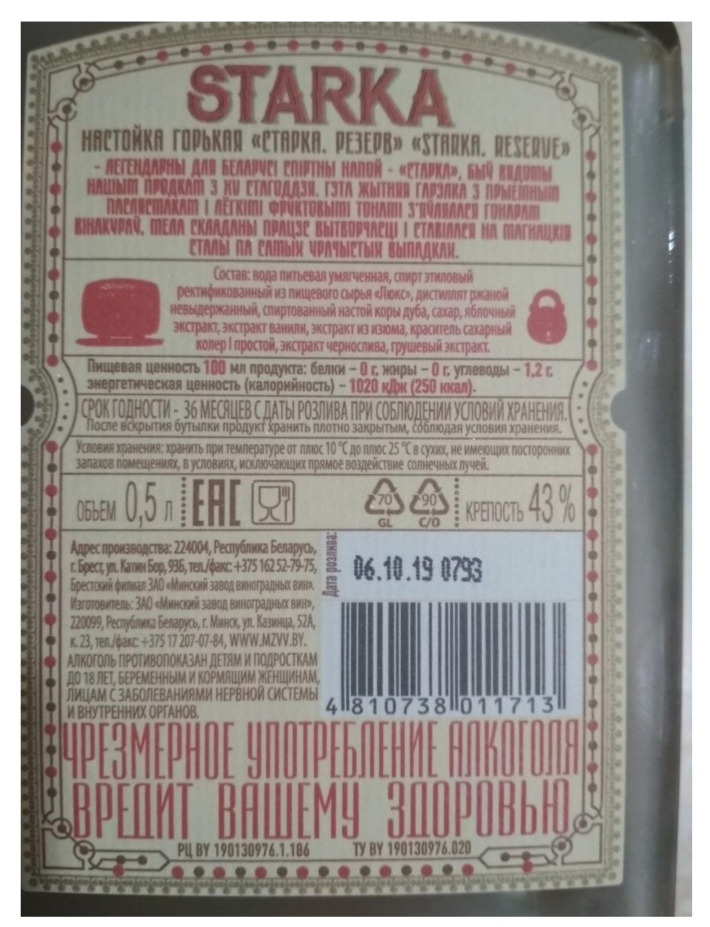
Alcohol label on a spirits bottle from Belarus. The health warning “Excessive alcohol consumption is harmful to your health” is featured in contrasting red capital letters at the bottom of the label and occupies about 1/6 of the label’s surface.

**Figure 4 ijerph-17-08205-f004:**
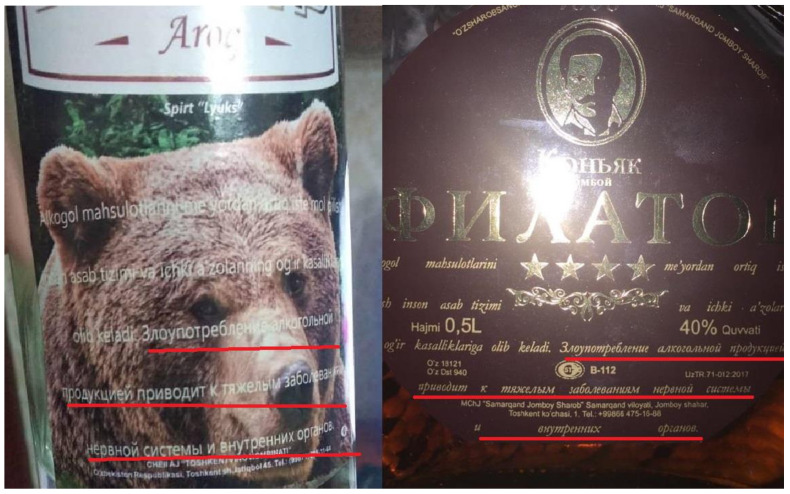
Alcohol labels on two spirits bottles from Uzbekistan. The health warnings with a “40% of label size” requirement are featured in Russian and Uzbek languages but do not stand out as they merge seamlessly with the background. For better readability, the Russian text of the health warning “Злоупотребление алкогольной продукцией приводит к тяжелым заболеваниям нервной системы и внутренних органов” (“Alcohol abuse leads to severe diseases of internal organs and nervous system”) is underlined in red.

**Figure 5 ijerph-17-08205-f005:**
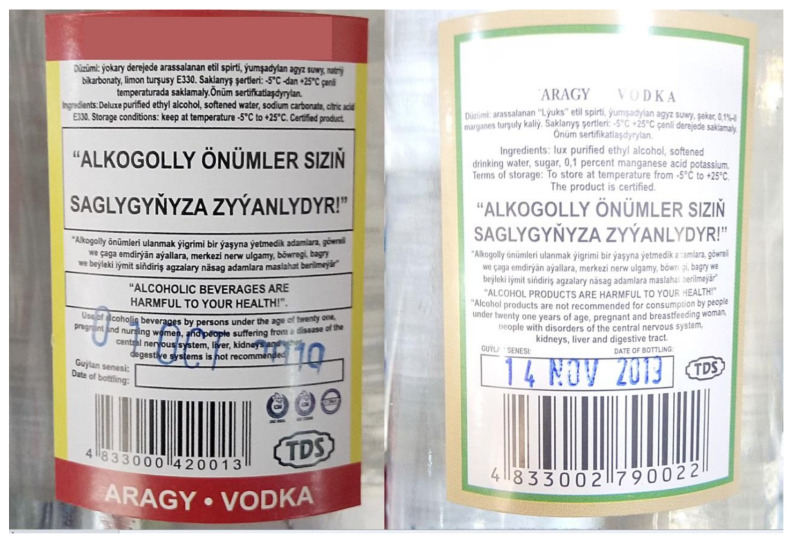
Alcohol labels on two spirits bottles from Turkmenistan. Left: the health warning is presented in Turkmen and English in a box with thick borders, which gives a visual separation between the health warning and the background. Right: The same health warning as on the left but without a box.

**Table 1 ijerph-17-08205-t001:** Overview of regulations on health warnings for alcohol advertisements and labels on containers in the CIS, including the specific messages used. Countries with an asterisk are member states of the Eurasian Economic Union, where the Technical Regulation EAEU 047/2018 on Alcohol Safety enters into force by 2021, superseding any previous national regulations.

Country	Health Warning(s) in Alcohol Advertisements	Health Warning(s) on Alcohol Containers	Exact Provision on Health Warnings and Related Regulatory Document(s)
Armenia*	No.	No.	-
Azerbaijan	Yes. Advertisements for alcoholic beverages with more than 5% alcohol by volume (ABV) must be accompanied by a warning about the harms regarding excessive consumption of the beverage to health. No specific content or form is defined.	No. Attempts to introduce pictorial health warnings were unsuccessful.	-
Belarus*	Yes. Advertisements for alcoholic beverages must be accompanied by the following health warning, in clear letters contrasting with the background, at the bottom of the advertisement and along its entire length, and occupy at least 10% of the advertising space: “Excessive alcohol consumption is harmful to your health.”	Yes. The same rules as for alcohol advertisements apply for the health warning: “Excessive alcohol consumption is harmful to your health.” Additionally, age and other consumption restrictions need to be featured, as follows: “Alcohol is contraindicated for children and youth under 18 years of age, pregnant and breastfeeding women, people with disorders of the central nervous system and internal organs.”	Health warnings and other information must be provided following additional technical regulations for beer, wine, and spirits. A recent letter from the Ministry of the Republic of Belarus “On the issue of placement on labels and in advertising of alcoholic beverages and beer phrases that alcohol consumption is harmful to health” from August 2019 states that health warnings are implemented in Belarus following national and international provisions. Source: [24].
Kazakhstan*	Total ban on alcohol and tobacco (including electronic delivery systems and hookahs) advertising, including for trademarks and elements thereof.	Yes. All alcoholic beverages, except for beer, must feature information on the harms and contraindications of alcohol consumption, as follows: “Excessive alcohol consumption is harmful to your health. Alcohol is contraindicated for individuals under 21 years of age, pregnant and breastfeeding women, people with disorders of the central nervous system, kidneys, liver and digestive organs.”	Health information has to be provided in accordance with the Resolution No. 1081 “On approving the ‘Technical regulations on requirements for the safety of alcohol products’.” Source: [25]. This legislation was repealed in May 2019 since most of the producers were already following the EAEU regulations.
Kyrgyzstan*	Yes. Alcohol advertisements should be accompanied by text warnings containing information on the harms of alcohol consumption and the legal ban on the sale of alcohol to minors. Both messages must be featured in the same size and font size, and together they must occupy at least 10% of the advertising space. The color of the warning text should be in contrast to the color of the warning’s background.	No. Attempts to introduce pictorial health warnings and health warnings with at least 40% of the label’s size were unsuccessful.	-
Moldova	No.	Yes. A pictogram indicating that alcohol should not be used during pregnancy and a pictogram indicating that alcohol should only be used by people of 18 years and older.	The main law on alcohol, law No. 1100-XIV “About production and turnover of ethyl alcohol and alcoholic products,” imposes the rules on labeling. Source: [26].
Russia*	Yes. Advertisements of alcoholic beverages must be accompanied by the following health warning, in clear letters contrasting with the background, at the bottom of the advertisement and along its entire length, and occupy at least 10% of the advertising space: “Excessive alcohol consumption is harmful to your health.”	Yes. Same rules as for the alcohol advertisements apply for the health warning: “Excessive alcohol consumption is harmful to your health.” Additionally, age and other consumption restrictions need to be featured: “Alcohol is contraindicated by children and youth under 18 years of age, pregnant and breastfeeding women, people with disorders of the central nervous system, kidneys, liver and digestive organs.” No size is specified by the ministry order. Attempts to introduce pictorial health warnings were unsuccessful.	The main law on alcohol, federal law No. 171, mandates the provide information on the “health harms of consuming alcoholic products” to consumers as part of the product labeling rules. The health warning is “communicated to consumers in the manner established by the Government of the Russian Federation.” The Order of the Ministry of Health and Social Development of the Russian Federation (Ministry of Health and Social Development of Russia) of 19 January 2007 N 49, Moscow “On the approval of a warning inscription on the consumer container of a unit of alcoholic beverages on contraindications to the use of alcoholic beverages” imposes the specific health warning. Source: [27].
Tajikistan	Total ban on alcohol and tobacco advertising.	No.	-
Turkmenistan	Total ban on alcohol and tobacco advertising.	Yes. A health warning in black capital letters on a white background in bold, clear, easy-to-read font, occupying at least 20% of the size of the label for domestically produced alcoholic beverages. “Alcohol is harmful to your health!” Additionally, age and other consumption restrictions need to be featured: “Alcohol is contraindicated for individuals under 21 years of age, pregnant and breastfeeding women, people with disorders of the central nervous system, kidneys, liver and digestive organs.”	The main law that regulates alcohol, law No. 99-VI “On the prevention of the harmful effects of alcohol.” Source: [28].
Uzbekistan	Total ban on alcohol and tobacco advertising.	Yes. A health warning with at least 40% of the size of the label in Uzbek and Russian. The following health warning has to be featured on the front label: “Alcohol abuse leads to severe diseases of internal organs and nervous system.” The legislation also allows for a graphical health warning.	The Annex to the Order of the Director-General of the Uzbek Agency for Standardization, Metrology and Certification dated 1 June 2016 No. 315 “Changes to the procedure for marking with the mark of conformity for alcoholic beverages produced by enterprises on the territory of the Republic of Uzbekistan” mandates “a medical warning occupying at least 40% of the label, in the form of a text and (or) a picture.” Source: [29].

**Table 2 ijerph-17-08205-t002:** Comparison between drafts and the adopted version of the EAEU technical regulation.

**Health Warning(s) as Suggested in the First Draft Developed by the Russian Federation**	**Health Warning(s) as Suggested in the Draft and Disseminated by the World Trade Organization**	**Health Warning as Adopted in the Final Technical Regulation EAEU 047/2018**
“Alcohol is harmful to your health.” The warning must be at least 20% of the label size. Black capital letters on a white background, in bold, clear, easy-to-read font of the largest possible size, with 37 line spacing not exceeding the font height. \ Information may be applied in any way that allows it to be clearly and easily read by a person with normal visual acuity in good lighting. The font size is at least 8 points. Inscriptions, signs, and symbols must have a contrasting background on which the marking is placed.	“Excessive alcohol consumption is harmful to your health.” The warning must be at least 20% of the label size. Contrasting capital letters in an easy-to-read font of the largest possible size. \ Information is applied in any way that allows it to be clearly and easily read by a person with normal visual acuity in good lighting. Inscriptions, signs, and symbols must have a contrasting background on which the marking is placed. Additional message: “Not recommended to be used by persons under the age of 21, pregnant and breastfeeding women, persons with diseases of the central nervous system and inner organs.”	“Excessive alcohol consumption is harmful to your health.” The warning must be at least 10% of the label size. Contrasting capital letters in an easy-to-read font of the largest possible size \ Information is applied in any way that allows it to be clearly and easily read by a person with normal visual acuity in good lighting. Inscriptions, signs, and symbols must have a contrasting background on which the marking is placed. Additional message: “Consumption not recommended for persons under the age of 18, pregnant and breastfeeding women, persons with diseases of the central nervous system and inner organs.” For products sold in Kazakhstan: “Alcohol is contraindicated for persons under 21 years of age, pregnant and breastfeeding women, persons with diseases of the central nervous system, kidneys, liver and digestive organs.”

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
