# Peer review of "Implementing Health Warnings on Alcoholic Beverages: On the Leading Role of Countries of the Commonwealth of Independent States"

_ijerph, 2020, doi:10.3390/ijerph17218205_

Round 1

Reviewer 1 Report

Dear authors, this is an excellent piece of research and exceptionally interesting and important. You shed light on the labelling regulation of a bloc of countries which have hitherto received very little attention. In that way, the study makes a highly significant contribution. The study is very detailed and helpful to those seeking to understand the trends in alcohol labelling around the world. I have made detailed comments in the attached document. I hope the comments are helpful to you. 

Author Response

Reviewer 1

Dear authors, this is an excellent piece of research and exceptionally interesting and important. You shed light on the labelling regulation of a bloc of countries which have hitherto received very little attention. In that way, the study makes a highly significant contribution. The study is very detailed and helpful to those seeking to understand the trends in alcohol labelling around the world. I have made detailed comments in the attached document. I hope the comments are helpful to you. 

  1. Introduction

  • Excellent introduction – very clear context for the paper and outline of what the paper offers. My only substantive comment is that the introduction pus the focus on alcohol warnings on labels but the results section looks at warnings on labels and It might be best to just focus on warnings on labels. But if you choose to do warnings on ads, I think you need to set this up in the introduction and then separate out the two sets of regulation in the results and discussion sections. There are real differences between warnings on ads and warnings on labels.

A1: We agree that there are big differences between the two and have therefore clarified in the Introduction that the article will focus mainly on alcohol labels (newly added lines 108-113). We have also added a section specifically on the issue of alcohol health warnings and advertisements as part of the discussion section (ll.304-315).

  • Page 2, first sentence: ‘as of today’: it would be best to include a month and year, rather than say ‘today’
  • Page 2: second para: should read ‘only recently have there been empirical studies’
  • Page 2: ‘according’ does not sound like quite the right word. Maybe ‘associated’?
  • Page 2: ‘the alcohol consumers’ change to ‘alcohol consumers’
  • Page 3: use of word ‘according’ – maybe change to ‘associated’?
  • Page 3: ‘regulatory framework point of departure’ change to ‘regulatory framework as a point of departure’.

A2: All these changes have been made.

  1. Materials and methods

  • Page 3: ‘legislations’ – this is usually written as ‘legislation’ whether singular or plural (in least in common law countries)
  • Page 3: the word used is ‘advertisement’. This might be correct and should be kept if that is the case, but ‘advertising’ sounds
  • Page 3: ‘this phrases’ change to ‘these phrases’
  • Page 3: extra ]

  1. Results
  • I found the results section a little confusing. The confusion for me comes from combining, in some places, the discussion of warnings on ads and warnings on labels. I think this needs to be very clearly separated
  • Page 3: I think it is best I the first senetence to separate out those countries which have national regulation of warnings and labels, national regulation of warnings and advertisements, and those countries to be subject to the technical regulation.
  • If you do separate out labels and advertisements, then the numbers need to change in the first I am also not sure that it is fully correct as it stands now. Eg, Armenia requires warnings on neither labels or ads.

A3: We have restructured the results section, adding subheadings and focusing the analysis more on the alcohol labels and the respective country examples. The difference between national and supranational regulatory frameworks should be now clearer as we provided more details on the EAEU regulation (ll. 1791-181). Because it seems very tricky to count the countries given the different dimensions of analysis (national/supranational and labels/ads + the total marketing ban in some countries), we removed the numerical information from the text to avoid confusion and referrer the readers to Table 1 now .

  • Page 3: Maybe ‘regulatory documents and food and alcohol labelling’ could be ‘regulatory standards for food and alcohol labelling’?
  • Page 3: I was not sure that ‘international’ was the correct way to describe the Eurasion Economic Union. It might be best to say ‘regional’ or ‘supra- national’. The latter is how EU regulation is

A4: All these changes have been made.

  • Can you add some information about when the EAEU technical regulation comes into force? Is it a specific date or is at some trigger point? Is it the same date for all members? Is it acceptable for members to transition to the new system before it comes into force for all of the members?
  • Also, can you explain how it works that the EAEU technical regulation applies int he member states? Does the treaty or agreement which makes countries members of the EAEU state that laws of the EAEU automatically apply in the member countries or do the member countries need to pass a law to bring the EAEU technical regulations into their national law? If the former, do states need to repeal their national laws for the EAEU law to operate? I ask because of the note in the table about Kazakhstan that the country repealed its national law because most producers were following the technical regulation already. Were they doing so voluntarily?
  • Is the technical regulation more onerous in all instances compared to the national laws of members? Also, see point below about the font

A5: This information is now given in the introduction (ll. 175-181) and can be also found partly in the discussion section dedicated to the EAEU regulation (ll. 263-296).

  • Figure 1: repeat of word ‘that’
  • Figure 1: I found this a little
    • Blue is the five countries which will be subject to the EAEU technical regulation. Red is presented as the ‘other CIS countries’ with national legislation but the box on figure 1 lists all of the 10 CIS countries and not just the ‘other five’.
    • Also, the red seems to cover countries with health warnings on ads

and/or labels. This puts different forms of regulation together.

  • Psge 4: missing closed brackets after (21).
  • Page 4, missing comma after ingredients list

A6: All these changes have been made.

  • Page: I would suggest adding a final sentence after ‘in line with the regulation’: ‘This means that products sold in [name of five EAEU countries] must bear these labeling features from [X] date. For [y] countries, this means that their current national systems of alcohol labeling regulation will not apply to the extent of any inconsistency the standard mandated by the technical regulation.’ Is this last point correct that if the countries have labeling laws which do not overlap with the technical regulation, these labeling laws will still apply but the TR will ‘knock out’ any national regulation to the extent that they cover the same ‘field’. What about if the national regulation provides for alcohol label warnings in a CERTAIN FONT, but the technical regulation does not provide for any particular FONT. Does the national law still apply in relation to the FONT or is this national requirement also ousted? Can national governments add further requirements which go beyond the technical regulation (ie, the technical regulation becomes a floor and not a ceiling)? These comments are driving at understanding in more detail how the bodies of law – national and supranational/regional fit
  • Page 9, first sentence: the numbers presented there do not match the numbers presented in the first sentence of the results section on page

A7: We have now clarified how the EAEU regulation will supersede national regulations in January 2021(ll. 175-181). Furthermore, we argue in the discussion section (ll.366-375) that the TR EAEU 047/2018 can and should be seen as the floor and not the ceiling in relation to graphical health warnings. Although Member States may no longer change the exact wording of the health message, they are free to add graphical health warnings and other elements as long as the requirements and the according regulatory act do not directly contradict the TR EAEU 047/2018. That is, at least, our current understanding of the regulatory framework. Because the technical regulation is quite new, there is a lot of uncertainty around it and the national and supranational negotiation and harmonization process is still ongoing. For instance, Russian beer brewers have just requested the Russian Ministry of Economic Development to postpone the entering into force of the technical regulations for a year because of its legal definitions of beer and beer-based beverages (https://regnum.ru/news/economy/3067393.html). We have tried to reflect the novelty of this regulation and the ongoing harmonization process in the manuscript, which again, should be seen as a fist entry point to the discussion.

  • Figure 2: is it possible to make the text on the bottles clearer to read or maybe do a frame or outline around the text on the right hand bottle which contains the new warning
  • Figure 2 note: when you say that the ‘excessive alcohol consumption is harmful to your health’ statement is likely elsewhere on the bottle, where would it be if it were not on the label? Is there another label?
  • Figure 2: is that the front or back label where the warning appears? Could you also check how large the font is relative to the volume/size of the container? It is a good piece of information for readers about how producers interpret ‘largest possible size’ in the technical regulation. Also, (and I am sorry I can’t read the text), do you think that the producer of the product might have been squeezing lots of different types of information onto the back label to try to make the warnings as small as possible? The size of the text on the second half of the bottle contrasts with that on the top half of the bottle where the producer describes their product. [I see this issue is at least partly addressed below on page 10 where you discuss the lack of presentation requirements in the technical ]

A8: The red frames for the health messages (Russian and Armenian) were added and it was clarified that Figure 2 depicts not the health warning as such, but the recommendatory message on contraindications only. We have also clarified elsewhere in the text (ll. 166-167) that per EAEU technical regulation, the health warning “Excessive alcohol consumption is harmful to your health” can be featured not only on the front label and that the size requirements apply only to this messages, whereas the provisions for the second “recommendatory message” on health information are not so strict (ll. 162-170). We hope that this provides more clarify when looking at different country examples.

  • Page 10, first sentence: maybe this should be reworded, ‘The same is true for Kyrgyzstan that has not got national legislation on alcohol health warnings, but has introduced them in accordance with EAEU ’
  • Page 10: I think word ‘front’ in second para should be ‘font’
  • Page 10: ‘not fully in with’ should be ‘not fully in line with’
  • Page 10: Belarus example – is this product purportedly complying with the national law or the technical regulation?
  • Page 12: ‘systematically’ change to ‘systematic’

A9: All these changes have been made.

  • Whole section but page 13 in particular: I think that some sub-headings might be good in this section but especially on page 13 where you present the results of your study of the differences between the draft and final versions of the technical
  • This is just a suggestion and it will be for you to consider whether this would fully work but I wonder about breaking the first part of the results section (where you discuss the labeling laws in different countries) into four groups (or maybe two – EAEU and non-EAEU membes):
    • countries which are members of the EAEU and have no national label requirements;
    • countries are members of the EAEU and have national label requirements;
    • countries which are not members of the EAEU and have label requirements;
    • countries which are not members of the EAEU and have no label requirements.

I started to get a little confused about which countries are subject to the technical regulation and which are not. It seems pertinent to be separately analyzing EAEU members, compared to non-members. From my perspective, one issue is whether the technical regulation sets more onerous requirements than the national legislation in all respects?

A10: We have restructured the results section (see comment above).

  • Page 14; should it be ‘soon thereafter’?
  • Page 14: could you explain what the ‘according resolution’ is?
  • Page 14: ‘since it may complicate and the placement of’: extra ‘and’?
  • Page: should ‘front’ be ‘font’?
  • Page 14: you may need to say a little about how these technical regulation get negotiated and ‘passed’ (?) in the EAEU. Does one member propose a resolution

for a technical regulation? Is there a consultation process? Do members vote on the resolution? Were all members in support of the draft of this technical regulation? Or was it Russia proposing the resolution for the technical regulation and other members were opposing it outright or seeking modifications? I don’t know about the EAEU processes but I was not quite sure whether the EAEU members negotiate about the terms of a proposed technical regulation through the WTO or whether they have their own forums for negotiation and other WTO members use the WTO forums for their input?

A11: All editing changes have been made. We do not feel qualified to give more detailed information on the processes within the EAEU and WTO as part of this contribution as this would require a more in-depth analysis and interpretation of internal documents that goes beyond the scope of the current review.

THE PAGE NUMBERS OF THE MANUSCRIPT END AT 14 AND THEN START AT 1 AGAIN WITH LINE NUMBERS DOWN THE SIDE OF THE PAGES

  • NEW PAGE 2: ‘author report’ should ‘author reports’
  • NEW PAGE 2: spelling error: eexcessive
  • NEW PAGE 2: suggested change to expression: ‘the two regulations that are most aligned with the WHO-suggested principles of alcohol labeling, in terms of the design as well as the content of the health warnings’
  • NEW PAGE 2: maybe change the word ‘term’ to ‘name of an alcohol abuse disorder’

A12: All these changes have been made.

  • NEW PAGE 3: you take up on this page the issue I have raised above about what ‘space’ the new technical regulation leaves for members to add further labeling requirements. I raised the example of font size and you give the example of graphics. Maybe you could address my comment about fonts (or other design features) from above down here?
  • NEW PAGE 3: with the discussion of the WTO, the TBT Committee does not ‘test’ or check the labeling regulation for consistency with the Agreement on Technical Barriers to Trade. WTO members may raise concerns about new measures there but it is only in formal dispute settlement that the labeling regulation gets tested or checked for its conformity with WTO Thailand has been the only WTO member to have its proposal for graphic health warnings discussed in the TBT Committee. It has not proceeded with these warnings and it seems possible that a contributing factor to not proceed with the warnings was the pressure placed on Thailand in the TBT Committee. I have a few papers on the issues which might be helpful to you:
    • O’BRIEN, P., and MITCHELL, A.D., ‘On the Bottle: Health Information, Alcohol Labelling and the WTO Technical Barriers to Trade Committee’ (2018) 18 Queensland University of Technology 124-55 (open access: https://lr.law.qut.edu.au/article/view/732).
    • O’BRIEN, P., ‘Australia’s Double Standard on Thailand’s Alcohol Warning Labels’ (2013) 32 Drug and Alcohol Review 5-10.

A13: We have addressed this as part of the discussion section (ll. 368-379) and in the conclusion (ll.417-422).

  1. Conclusion
    • NEW PAGE 3: use of descriptor ‘international’: see above whether regional or supra- national might be
    • NEW PAGE 3: change ‘legislations’ to ‘legislation’
    • NEW PAGE 3: I would suggest making the argument clearer that the public health guidance about font size and other design features needs to be in the legislation
    • NEW PAGE      3:     ‘industry    interference   who’     –     change    who    to     which 8
    • NEW PAGE 3: see above re WTO – the WTO does not need to ‘approve’ the new regulations. They will stand unless they are formally challenged and found to be

inconsistent with WTO law or until the state chooses to amend them or water them down.

  • NEW PAGE 4: grammar: harmonization ‘was’ not ‘were’

A14: All these changes have been made and the issue with the WTO addressed as part of the conclusion (see comment above).

Reviewer 2 Report

An informative paper summarising the implementation of alcohol warning labels in countries of the Commonwealth of Independent States, which serves as an example to the rest of the world. Hopefully, this paper will stimulate discussion of this important issue and encourage adoption by more countries. The manuscript was well written and only minor editing is needed.

Author Response

Reviewer 2

An informative paper summarising the implementation of alcohol warning labels in countries of the Commonwealth of Independent States, which serves as an example to the rest of the world. Hopefully, this paper will stimulate discussion of this important issue and encourage adoption by more countries. The manuscript was well written and only minor editing is needed.

A1: The manuscript has been edited.

Reviewer 3 Report

Dear authors,

A very good article fully explaining the issue. The conclusions are very interesting for readers and are supported by your results.

- The formatting is not very thorough throughout the whole document: caption for figure on page 12 is wrong. Given the page numbering is not correct, I will refer to the page number as in the PDF file.

- Page 2, second paragraph: maybe change to "only recently have there been" and later in the paragraph "not yet" for correct grammar;

- Page 4, first sentence: review the sentence, it is difficult to understand what you mean and it is missing a parenthesis.

- Please review figure 1: difficult to understand the map with the colours used, for example the red is very similar to other colours used;

- Page 9, line 1: "have" instead of "has";

- Page 10 line 1: "The same...", "already had" instead of "had already". Second paragraph in page 10 "have" instead of "has". In this paragraph when you mention "front" do you mean "font"? If so change throughout the text. Third paragraph first line "were" instead of "was"; and in point 5) maybe you could delete "of" and "for" so it reads "its size the same as..."; last paragraph second line "is also" instead of are.

- Page 11, first paragraph: do you mean graphic instead of graphical? If so change throughout the text

-Page 12, first paragraph "systematic" instead of systematically;

-Page 15: line 6 "soon made public"; line 13 "was" instead of "were"; line 14 wrong construction, please review what you meant; line 36: "aware of"; sentence starting in line 59 in page 15 is not gramatically correct; Line 61 "excessive"; 

-Page 17 line 117 "which" instead of "who"

-Page 18 line 128 "was" intead of "were"

Would have been interesting to see you write about recommendations based on your work: for example how can your work be used by public health practitioners and authorities to counteract industry's influence?

Author Response

Reviewer 3

A very good article fully explaining the issue. The conclusions are very interesting for readers and are supported by your results.

- The formatting is not very thorough throughout the whole document: caption for figure on page 12 is wrong. Given the page numbering is not correct, I will refer to the page number as in the PDF file.

- Page 2, second paragraph: maybe change to "only recently have there been" and later in the paragraph "not yet" for correct grammar;

- Page 4, first sentence: review the sentence, it is difficult to understand what you mean and it is missing a parenthesis.

- Please review figure 1: difficult to understand the map with the colours used, for example the red is very similar to other colours used;

- Page 9, line 1: "have" instead of "has";

- Page 10 line 1: "The same...", "already had" instead of "had already". Second paragraph in page 10 "have" instead of "has". In this paragraph when you mention "front" do you mean "font"? If so change throughout the text. Third paragraph first line "were" instead of "was"; and in point 5) maybe you could delete "of" and "for" so it reads "its size the same as..."; last paragraph second line "is also" instead of are.

- Page 11, first paragraph: do you mean graphic instead of graphical? If so change throughout the text

-Page 12, first paragraph "systematic" instead of systematically;

-Page 15: line 6 "soon made public"; line 13 "was" instead of "were"; line 14 wrong construction, please review what you meant; line 36: "aware of"; sentence starting in line 59 in page 15 is not gramatically correct; Line 61 "excessive"; 

-Page 17 line 117 "which" instead of "who"

-Page 18 line 128 "was" intead of "were"

Would have been interesting to see you write about recommendations based on your work: for example how can your work be used by public health practitioners and authorities to counteract industry's influence?

A1: The manuscript has been edited and the specific recommendations were added to the conclusion (ll. 436-447).

Reviewer 4 Report

The article is novel, interesting and well written. However, it provokes several questions which are not answered/ discussed in the text:

  1. Why “provision of consumer information” is not discussed in the text? From a public health perspective, is this less relevant than health warnings?
  2. What is the historical and social background of discussed regulations in the 10 countries of the Commonwealth of Independent States (CIS)? When were they introduced? What is the prevalence of alcohol consumption and related problems in the CIS countries? How the differences across countries in the present regulations can be explained?
  3. Is the overview of regulations of health warnings on alcohol advertisements and labels on containers in the CIS presented in the table 1 detailed enough? E.g. only for Azerbaijan the definition of alcoholic beverages is provided (“with more than 5% ABV”). What is meant by “alcohol” in other cases? It looks like in just one in 10 countries (Uzbekistan) the regulations concern “alcoholic beverages produced by enterprises on the territory of the Republic” while in all other member states the same labeling is required for domestic and imported products. Is this true?

The figures showing examples of alcohol labels are not clear, even for Russian speakers. In the figures 2 and 3 it might be helpful to cite the health warnings not only in English translation but also in the original language, or to indicate graphically their location in the picture.  In the figure 4 (in the text presented as figure 1), Right – the warning text is not visible at all. Is it on purpose?   

My last comment concerns the lack of clearly formulated conclusions that result from the conducted analysis for other countries / regions. It would be helpful to see what is the lesson we should learn from the CIS countries experiences.

Author Response

Reviewer 4

The article is novel, interesting and well written. However, it provokes several questions which are not answered/ discussed in the text:

  1. Why “provision of consumer information” is not discussed in the text? From a public health perspective, is this less relevant than health warnings?

A1: We believe that the referenced Health Evidence Synthesis report on alcohol labeling from the WHO deals extensively with this question and that the present contribution could not add much more to this analysis. We have now clarified in the introduction section (ll. 109-113) and throughout the entire manuscript that the main scope of the present contribution is the analysis of the implementation of health warnings on alcohol containers in the CIS as well as the discussion of country examples to highlight some gaps and shortfalls in the current provisions.

  1. What is the historical and social background of discussed regulations in the 10 countries of the Commonwealth of Independent States (CIS)? When were they introduced? What is the prevalence of alcohol consumption and related problems in the CIS countries? How the differences across countries in the present regulations can be explained?

A2: As we have said in the discussion section, there is almost no literature on the topic of health warnings in Russian language and in the CIS, which is why we cannot address this query. We can only assume that some form of health warning regulations were already in place in the Soviet Union, possibly introduced as part of the several Soviet anti-alcohol campaigns, but we do not have any materials to support this assumption. We tried to reflect this in the discussion section (ll. 316-320).

  1. Is the overview of regulations of health warnings on alcohol advertisements and labels on containers in the CIS presented in the table 1 detailed enough? E.g. only for Azerbaijan the definition of alcoholic beverages is provided (“with more than 5% ABV”). What is meant by “alcohol” in other cases? It looks like in just one in 10 countries (Uzbekistan) the regulations concern “alcoholic beverages produced by enterprises on the territory of the Republic” while in all other member states the same labeling is required for domestic and imported products. Is this true?

A3: Table 1 provides only the information that we could identify as part of the national and supranational regulations and the available documents. We did not check for the national definitions of alcohol and alcoholic beverages as we felt that this would go beyond the scope of this exploratory review.

The figures showing examples of alcohol labels are not clear, even for Russian speakers. In the figures 2 and 3 it might be helpful to cite the health warnings not only in English translation but also in the original language, or to indicate graphically their location in the picture.  In the figure 4 (in the text presented as figure 1), Right – the warning text is not visible at all. Is it on purpose?   

A4: We have altered this Figure for better readability. The fact that the health warning text appeared to be not visible is yet again underlines the point that we made in the manuscript.

My last comment concerns the lack of clearly formulated conclusions that result from the conducted analysis for other countries / regions. It would be helpful to see what is the lesson we should learn from the CIS countries experiences.

A5: We have added the specific recommendations to the conclusion (ll. 436-447).